# Curcuminoid Co-Loading Platinum Heparin-Poloxamer P403 Nanogel Increasing Effectiveness in Antitumor Activity

**DOI:** 10.3390/gels8010059

**Published:** 2022-01-14

**Authors:** Ngoc The Nguyen, Quynh Anh Bui, Hoang Huong Nhu Nguyen, Tien Thanh Nguyen, Khanh Linh Ly, Ha Le Bao Tran, Vu Nguyen Doan, Tran Thi Yen Nhi, Ngoc Hoa Nguyen, Ngoc Hao Nguyen, Ngoc Quyen Tran, Dinh Trung Nguyen

**Affiliations:** 1Faculty of Medicine-Pharmacy, Tra Vinh University, Tra Vinh City 87000, Vietnam; nguyentienthanh@tvu.edu.vn (T.T.N.); lykhanhlinh@tvu.edu.vn (K.L.L.); 2Institute of Applied Materials Science, Vietnam Academy of Science and Technology, Ho Chi Minh City 71500, Vietnam; quynhanhbui011@gmail.com (Q.A.B.); haonguyen101297@gmail.com (N.H.N.); tnquyen979@gmail.com (N.Q.T.); 3Faculty of Biology and Biotechnology, University of Science—Vietnam National University, Ho Chi Minh City 72700, Vietnam; huongnhu2608@gmail.com (H.H.N.N.); tlbha@hcmus.edu.vn (H.L.B.T.); dnvu@hcmus.edu.vn (V.N.D.); 4Graduate University of Science and Technology, Vietnam Academy of Science and Technology, Ho Chi Minh City 71500, Vietnam; ttynhi@ntt.edu.vn; 5Institute of Environmental Technology and Sustainable Development, Nguyen Tat Thanh University, Ho Chi Minh City 72800, Vietnam; 6German Vietnamese Technology Center, HCMC University of Food Industry, Ho Chi Minh City 72000, Vietnam; hoann@hufi.edu.vn

**Keywords:** poloxamer P403, multi-drug delivery, heparin, MCF-7, nanogel

## Abstract

Nanosized multi-drug delivery systems provide synergistic effects between drugs and bioactive compounds, resulting in increased overall efficiency and restricted side effects compared to conventional single-drug chemotherapy. In this study, we develop an amphiphilic heparin-poloxamer P403 (HP403) nanogel that could effectively co-load curcuminoid (Cur) and cisplatin hydrate (CisOH) (HP403@CisOH@Cur) via two loading mechanisms. The HP403 nanogels and HP403@CisOH@Cur nanogels were closely analyzed with ^1^H-NMR spectroscopy, FT-IR spectroscopy, TEM, and DLS, exhibiting high stability in spherical forms. In drug release profiles, accelerated behavior of Cur and CisOH at pH 5.5 compared with neutral pH was observed, suggesting effective delivery of the compounds in tumor sites. In vitro studies showed high antitumor activity of HP403@CisOH@Cur nanogels, while in vivo assays showed that the dual-drug platform prolonged the survival time of mice and prevented tail necrosis. In summary, HP403@CisOH@Cur offers an intriguing strategy to achieve the cisplatin and curcumin synergistic effect in a well-designed delivery platform that increases antitumor effectiveness and overcomes undesired consequences caused by cisplatin in breast cancer treatment.

## 1. Introduction

Cisplatin (Cis-diamminedichloroplatinum (II)) is a standard first-line treatment for human breast cancer [1,2,3,4,5,6]. It is one of the prime anticancer drugs for many types of solid tumors and the first platinum compound approved by the FDA for treating testicular and ovarian cancer since 1978 [3,4,5,6]. However, the toxicity of cisplatin has caused many side effects on the kidney, marrow failure, chronic neurotoxicity, etc., leading to drug resistance and limiting dosage during treatment. Moreover, the poor selectivity between normal and tumor tissue has hindered cisplatin’s prospect in clinical fields [6,7]. Several approaches have been developed to improve the therapeutic efficiency of cisplatin and overcome its limitations, such as novel platinum drugs, nanosized delivery systems, combination therapy treatment with other anticancer drugs, or phytochemical compounds (polyphenols, flavonoid, etc.) [2,4,6,7]. Among the regimens, therapeutic formulations of cisplatin and phytocompounds proved synergistic effects in cancer treatment and reduction of drug side effects, contributing to the expansion of its clinical usage [8,9,10].

Curcumin (Cur), a polyphenol derivative that originated in turmeric rhizome (*Curcuma longa* L.), has been shown to prevent skin cancer, stomach cancer, and bowel cancer in mice [11,12,13,14,15,16,17]. Besides having broad biological activity and excellent pharmacokinetics, curcumin is also non-toxic to small and large animals, even at high concentrations [18,19,20,21]. More interestingly, recent reports have revealed that Cur can create a synergistic effect with Cis, enhance tumor-proliferative inhibition and control side effects caused by mono-cisplatin regimens [8,9,10,22,23,24,25,26].

However, Cis and Cus are both hydrophobic compounds and have poor oral bioavailability [2,11]. In this study, we have attempted to load Cis and Cur into amphiphilic drug nano-carriers. Ideally, such nanosystems allow specific drug targeting, improving efficacy, and minimizing systematic toxicity [27,28]. One candidate of interest is poloxamer, a group of amphiphilic and biocompatible polymers [29]. Poloxamer has the symmetric triblock copolymer structure PEO-PPO-PEO, consisting of hydrophilic PEO regions (poly (ethylene oxide)) and hydrophobic PPO regions (poly (propylene oxide)) [30,31]. The hydrophobic cores containing water-insoluble drugs are surrounded by the hydrophilic shell, thus increasing the solubility and permeability of loaded compounds [32]. Within this category, poloxamer P403 is a prospective material due to its high ratio of PPO (as compared to poloxamer P407, pluronic F127), low critical micelle concentration (CMC) value, and considerable internal space [33]. The poloxamer-based drug-carrier system can also achieve higher oral availability by inhibiting the drug efflux mediated by P-glycoprotein [34].

Currently, poloxamer has often been conjugated with several biocompatible polymers, such as chitosan, gelatin, heparin, polyacrylic, dendrimer, etc. [31,35,36]. It is because poloxamer micelles alone couldn’t provide a durable, long-term drug-release profile; some even exhibit low biocompatibility. The grafted copolymers serve as multifunctional nanogel platforms for delivering various bioactive molecules while amplifying their biological interactions [30,34,37,38,39,40]. Heparin-based platforms exhibited a prominent potential for clinical applications due to their high biocompatibility and inherent interactions with cells [30,34,41]. Heparin is readily used in chemotherapy due to its high binding affinity to angiogenic growth factors that tumor tissues often overexpressed [42,43].

In this work, amphiphilic heparin-conjugated poloxamer P403 (HP403) nanogel was developed as carrier systems for CisOH and Cur. The physicochemical properties of the nano-carriers—particle size, drug entrapment efficiencies (EE), drug delivery efficiency (DL), and drug release—were characterized. In vitro cytotoxicity and in vivo assays were investigated in human breast cancer cell lines (MCF-7 cells) and xenograft assay models, respectively. The result could partially clarify the synergistic effect of aquated cisplatin and Cur in a well-designed delivery platform in antitumor activity.

## 2. Results and Discussion

### 2.1. Characterizations of the Amphiphilic HP403 Copolymer

The characteristic resonance peaks of the p-nitrophenyl chloroformate (NPC)-activated P403 appear at 1.10 and 3.35–3.70 ppm in ^1^H-NMR (Figure 1A). The aromatic proton signals of NPC moiety resonate at 7.39 ppm (m) and 8.27 ppm (l). The activation of the NPC molecule and P403 was confirmed with a new signal at 4.45 ppm (f) [44,45]. The chemical shift from 4.45 ppm (f) to 4.22 ppm (k) occurs due to the NPC substitution by 3-amino-1-propanol, which indicates the successful synthesis of NPC-P403-OH. The ^1^H-NMR spectrum of aminated heparin H-DAB (Figure 1B) shows the resonance signal at 1.59–1.75 ppm (i) characterized for methylene proton (–CH_2_–) on DAB [46]. The combination of heparin and DAB resulted in a signal of –CH_2_–NH– at 2.88–3.03 ppm (j), which belongs to the binding site of heparin and DAB carboxylate [47]. There are also signals for heparin proton at 1.99 ppm and 3.22–5.33 ppm [29,48]. The evidence indicates the successful synthesis of H-DAB. Figure 1C shows the ^1^H-NMR spectrum of HP403. Characteristic signals of P403 were 1.09 ppm (a) and 3.67 ppm (b), corresponding to the resonance of the –CH_3_ proton in PPO and the OCH_2_–CH_2_O– proton in PEO. Proton signals of heparin appeared at 1.99 ppm and 3.22–5.32 ppm. Notably, the linkage between H-DAB and NPC-P403-OH resulted in the resonance of methylene protons (–CH_2_–CH_2_ and –CH_2_–NH) at 1.74 ppm (i) and 3.11 ppm (j). In addition, there was no signal of aromatic protons of NPC. The signal performing a direct bond between the methylene and carbonate group of NPC (–CH_2_–O–NPC) at 4.43 ppm (f) did not occur. The only signal recorded at 4.22 ppm (k) belongs to methylene protons of CH_2_–O– Ami linkage. This result proves that NPC moiety was substituted by H-DAB amine groups in the grafted process of two polymers.

### 2.2. Size Distribution and CMC Value of HP403

The TEM result in Figure 2 shows that the HP403 nanogel has spherical shapes with an average particle size of 61.4 ± 20.2 nm. The average particle size measured by DLS shown in Figure 2C ranged 94.12 ± 3.85 nm at 25 °C with PI = 0.39 ± 0.07, which indicated that the nanogel system has high stability (Figure 2D).

The CMC of the HP403 copolymer was 40.97 ± 4.04 ppm, measured with iodine and UV-Vis spectroscopy. CMC is a crucial parameter to the nanogel formation of amphiphilic copolymers, especially in drug-delivery applications. The assembling of micelles through hydrophobic interactions helps to balance and stabilize drug release [49]. Within the HP403 structure, P403 is responsible for micelle self-assembly. Insoluble drugs are contained in the PPO hydrophobic core of poloxamer. The complex is shielded by the hydrophilic regions consisting of PEO and heparin with hydrogen bonding or electrostatic interactions. This unique structure makes nanogels an ideal and highly stable drug delivery system [30,50].

### 2.3. Drug Entrapment Efficiency and Loading Efficiency of HP403

In this experiment, Cur was loaded into HP403 copolymer through hydrophobic interactions between Cur and PPO regions of P403. CisOH was complexed to carboxylate/sulfate groups on heparin. The structure and encapsulation efficiency of HP403 towards Cur and CisOH were determined through FT-IR, UV-Vis and ICP-MS spectroscopy (Nex-ION 2000, Perkin Elmer, Waltham, MA, USA).

FT-IR spectra in Figure 3 show the typical signals of both heparin and P403: peaks 2951.95 cm^−1^ and 2874.03 cm^−1^ belongs to the C-H valence oscillations of -CH_2_ and -CH_3_ groups on poloxamer P403; peaks 3434.99 cm^−1^ and 3285.97 cm^−1^ are the -OH and -NH valence vibrations on the Cis molecule, respectively [51]. A new absorption oscillation appeared in the HP403@CisOH spectrum at 1712.01 cm^−1^ (A). The signal indicated a complex formed between CisOH and the heparin carboxylate group, which shifted the C=O carboxylate signal from 1631.89 cm^−1^ to 1712.01 cm^−1^. Furthermore, two new infrared absorption bands in HP403@CisOHs spectrum at 1298.02 cm^−1^ (B) and 845.96 cm^−1^ (C) are symmetric valence vibration of S=O bond due to the complex formation sulfate group and a part of the sulfonate group on heparin with CisOH [52]. The FT-IR spectroscopy in Figure 3 demonstrates the successful synthesis of HP403@CisOH product within the simultaneous appearance of HP403, CisOH, and the complex of CisOH and anionic groups on heparin. Furthermore, the spectrum of HP403@CisOH@Cur contained characteristic signals of both HP403@CisOH and Cur (peak 1627.98 cm^−1^ for C=C aromatic ring, peak 1509.02 cm^−1^ for C=O and C=C) [53]. The result proves that Cur was loaded and encapsulated in the HP403@CisOH nanogel system, and the HP403@CisOH@Cur nanoparticles were successfully synthesized. The DL and EE results of HP403@CisOH@Cur were calculated from formulas (1) và (2) in Section 4.3 shows that HP403@CisOH could load up to 30.39% and 75.98% CisOH, respectively. While the HP403@CisOH@Cur system successfully encapsulated 22.3% (DL) and 55.75% (EE) of CisOH; and 4.4% (DL) and 88% (EE) for Cur. 

TEM images were used to evaluate the morphology of HP403@CisOH@Cur nanoparticles. Figure 4 shows that HP403@CisOH@Cur had an average particle size of 125.72 ± 18.01 nm. The DLS result of HP403@CisOH@Cur nanogel was 162.9 nm with PI = 0.27, indicate the sample has high stability. An increment in dual drug-loaded nanogels was recorded due to loading a high amount of Cur in the hydrophobic domains and platinum complex formation resulting in expanding its particle size.

### 2.4. In Vitro Drug-Release Assay and Drug-Release Kinetics

The release of CisOH and Cur from the nanogel was investigated at pH 5.5 and 7.4. Figure 5A shows that, in the first 12 h, about 37.91% and 42.15% of Cur were released at pH 7.4 and 5.5, respectively. There was no burst drug release at this early stage, indicating that Cur was not adsorbed on the surface of the nanogel and was completely enclosed within the hydrophobic structural region of the nanogel. This resulted in a slow and sustained release of curcumin during the first 12 h at both pH media. The rate of Cur release then gradually increased over the next 84 h; higher Cur release was observed at pH 5.5. Approximately 75.70% of Cur was released at pH 5.5 and 59.04% of Cur was released at pH 7.4 after 96 h. Figure 5 shows that CisOH and Cur are both released more efficiently at pH 5.5. The acidic environment helped to accelerate the release of drugs, especially CisOH. It was beneficial for the system’s activities because tumors usually have a lower pH than healthy tissue. This result also corresponded to some previous publications of platinum carrier systems [54,55]. Figure 5B CisOH was not completely released from the HP403@CisOH@Cur system after 96 h. The release rate of CisOH only reached 68% in pH 5.5 and 47.43% in pH 7.4, which lead to the conclusion that HP403@CisOH@Cur was a potential slow-release carrier system. The drug-releasing process was inhibited by the complex between Cis and heparin’s carboxylate/sulfate groups, as well as the conjugation between CisOH and the amine/amide groups of heparin [53,56]. In the blood circulation (pH 7.4), the bonds in the carrier system are stabilized to prevent unwanted toxicity and non-specific effects of the drug. Once the drug-loading systems reach tumor tissues, the drug is rapidly released by the reaction of HP403@CisOH@Cur in the acidic environment of the tumor.

Drug delivery systems aim to overcome the problems of retention, release, activation, localization, and drug targeting with precise timing, dosage, and location. Biodegradable polymers are used as drug transporters in vitro or in vivo. By tuning their structure and drug release, we can control the kinetics of those drug carriers. The drug release kinetics of HP403 nanogel have been studied to understand the release mechanism of CisOH and Cur in vitro as a premise for future in vivo tests. Experimental data on the release of Cur and CisOH were processed through Solver and Data Analysis programs. The k and R^2^ kinetic constants were shown in Table 1. Our results showed differences in the value of regression parameters R^2^. The R^2^ values of different models followed this order: Korsmeyer–Peppas > Higuchi > Zero-order > First-order. The release of CisOH and Cur from HP403@CisOH@Cur at pH 7.4 and pH 5.5 followed the Korsmeyer–Peppas model. The diffusion exponent (n) of the system ranges from 0.2056 to 0.4271, calculated from the Korsmeyer–Peppas kinetic equation. Table 1 showed n < 0.45, indicating that CisOH and Cur were released from HP403@CisOH@Cur following Fickian diffusion [57,58].

The changes in particle size and PDI of HP403@CisOH@Cur after redispersion in an aqueous medium for 96 h were shown in Figure 6. At 72 h, there was no significant change. The particle size was less than 250 nm, and the PDI was less than 0.5. The results indicated a narrow size distribution of HP403@CisOH@Cur after redispersion in an aqueous medium at 72 h. At 96 h, particle size and PDI increased to more than 300 nm and more than 0.8, respectively. Our results show that HP403@CisOH@Cur can maintain its structural stability in an aqueous medium for 72 h.

### 2.5. Cytotoxicity of HP403@CisOH@Cur Nanogels

The biocompatibility of the HP403 carrier system is evaluated via SRB staining on fibroblasts. The results show that the cell survival rate reached 90% at 48 h after exposure to 100 µg/mL HP403. Therefore, the HP403 carrier was biocompatible and non-toxic. In contrast, Cis was extremely lethal to normal cells, with IC_50_ = 0.61 ± 0.03 µg/mL. Many clinical studies have also reported that Cis attacks both cancer and healthy tissues, causing severe side effects to patients taking chemotherapy drugs. However, once the drug is loaded into nanogel carrier systems in CisOH form, the hydrolysis of the complex inside the nanogel causes a slow release of the cisplatin hydrate. This process can reduce the toxicity of the Cis and may prolong drug bioavailability.

The cytotoxicity assay of HP403@CisOH and HP403@CisOH@Cur on the breast cancer cell line MCF-7 was performed by SRB staining, as shown in Figure 7. After 48 h, drug-loaded HP403 nanogels exhibited high inhibition towards MCF-7 cells. At 10 μg/mL, the HP403@CisOH system has a cytotoxic percentage of 52.76 ± 2.53%, higher than 40.45 ± 1.3% of HP403@CisOH@Cur. The above results indicated that the CisOH and Cur combination could reduce the side effects of CisOH but diminish the cytotoxic potential towards MCF-7 cancer cells of the drug-carrying nanogel system. IC_50_ value of HP403@CisOH is 8.5 ± 0.7 µg/mL, which is smaller than 19.99 ± 0.89 µg/mL of HP403@CisOH@Cur.

The results prove that HP403@CisOH@Cur can provide a synergistic effect against the proliferation of breast cancer cell MCF-7. Cisplatin is a powerful chemotherapeutic agent which induces apoptosis primarily through binding to DNA and inhibiting DNA replication [22]. However, the mechanism is not tumor-specific, and therefore cisplatin often attacks normal tissues during therapy. In addition, Cis resistance was frequently observed throughout the cases, leading to clinical failure [6,7]. Curcumin is an antioxidant with significant anti-inflammatory and anti-neoplastic activities. Several clinical studies have shown that curcumin can attenuate undesirable toxicity caused by cisplatin and reverse cisplatin resistance in cancer cells, which eventually enhances the therapeutic effect [8,9,10,22,23,24,25,26]. Furthermore, curcumin is rapidly metabolized under physiological conditions; thus, curcumin encapsulation in nanoparticles may contribute to retaining its bioactivity [18,19,20,21].

### 2.6. Antitumor Effect of HP403@CisOH@Cur Nanogel on Xenografted Mouse Assay

Mice of uniform weight were used to ensure correlation between treatments. The formulations were injected through their tail veins. Mice body weight and tumor volume were recorded over 13 days. The results are shown in Figure 8 and Figure 9.

As shown in Figure 8, mice maintained a stable weight in control (NaCl 0.9%) and HP403 samples. In contrast, there was considerable body mass loss after injecting cisplatin, HP403@CisOH, and HP403@CisOH@Cur. In the cisplatin and HP403@CisOH formulations, there were four dead mice on day eight, which could indicate that the formulated platinum contents could be too high resulting in high toxicity with investigated animals. The same loaded platinum amount in HP403@CisOH@Cur was quite high, reaching 22.3% (DL) and 55.75% (EE), which led to a decrease in the nanogel’s drug-holding capacity. Moreover, nearly 60% CisOH was released rapidly in the first 12 h at pH 5.5. These reasons might be responsible for the significant body weight loss after 13 days. Although, there were no mice that died in the treatment regimen after thirteen days, which could partially indicate synergistic effect of Cur in the dual delivery system. Further studies should reduce the amount of CisOH loaded in HP403@CisOH@Cur to increase clinically potential application of the drug delivery system. In Figure 9, we continue to observe two opposite trends: tumor volume was reduced in cisplatin, HP403@CisOH, and HP403@CisOH@Cur treatments but increased in control (NaCl 0.9%) and HP403 samples. There were no significant differences between NaCl and HP403 samples during the 13 days of testing. Tumors in both cases expanded rapidly and increased by nearly 60% after the last day. In contrast, tumor reduction was observed in cisplatin, HP403@CisOH, and HP403@CisOH@Cur. On day eight, the tumor decreased by 80%, 78%, and 70% of the original volumes, respectively. Thus, it can be seen that HP403 had significant enhancing effects on the anti-cancer ability of cisplatin. Furthermore, mice injected with HP403@CisOH@Cur can survive to day 13 and achieved 90% tumor volume loss. It indicated the dual curcumin and cisplatin nanocarrier system was the most effective regimen in inhibiting cancer cells growth.

It was observed in Figure 10C that necrosis occured at the tail area of mice injected with Cisplatin. These may be due to cisplatin’s high toxicity and bioaccumulation in the injected sites, causing inflammation that leads to necrosis. In Figure 10D, the tail of mice treated with HP403@CisOH suffered from inflammation and blood congestion. In contrast, the tail appearance of HP403@CisOH@Cur-treated model (Figure 10E) was normal, similar to the control and HP403 samples. Our results correspond with previous studies that reported curcumin’s anti-inflammatory activity that contributed to suppressing the side effect [13,14,15].

Besides the above assays for tracking the effectiveness of treated models, histological staining is one of the gold standards to evaluate the biopsy of cancer tissue. Hematoxylin and eosin (HE) staining results could further clarify the density of cancer cells in tumor tissue. The HE stained images of the xenografted tumors before treatment (Figure 11A) and after treatment with NaCl 0.9% and HP403 (Figure 11B,C) show hyperproliferation of MCF-7 cancer cells inside tumor, indicated by their stained nuclei with dark purple color. The proliferation of MCF-7 cells was also confirmed by superoxide dismutase immunohistochemical staining of tumor (data not shown here). In Figure 11D–F, Cis, HP403@CisOH, and HP403@CisOH@Cur treatments inhibited cancer cell proliferation leading to decreased cell density, degraded cytoplasmic region, and shrunk or disappeared nuclei. Compared to Cis and HP403@CisOH-treated models (day 8), the HP403@CisOH@Cur-treated model (day 13) exposed a lesser cell density with acquired lighter purple color. These results could indicate that co-loaded Cur contributed to reduce side-effects of the Cis and CiOH-treated models and prolong the life span of the mice.

## 3. Conclusions

In this study, we have developed a biocompatible heparin-poloxamer P403 platform to efficiently deliver cisplatin (in hydrate form) and Cur via complexation and hydrophobic interaction. The particle size distribution of HP403 was below 100 nm by (TEM and DLS). CisOH and Cur were successfully loaded into HP403 nanogel with 22.3% and 4.4% efficiency, respectively. In vitro kinetic results of HP403@CisOH@Cur showed that CisOH and Cur were sustainably released from the nanogel via a diffusion mechanism. The in vitro studies indicated that HP403@CisOH@Cur might provide a synergistic effect against the proliferation of MCF-7 breast cancer cells. According to the in vivo anti-tumor assay using the xenografted mouse model, the HP403@CisOH@Cur nanogel could reduce the tumor volume by 90% without mice dying after being treated for 13 days. The preliminary results indicated that the nanocarrier system of curcumin and cisplatin were highly effective at inhibiting the growth of cancer cells and partially reducing side effect. A further investigation should be proposed with a reduced dosage of the loaded platinum content to significantly reduce drug toxicity and exploit the effects of a dual drug delivery system in cancer treatment.

## 4. Materials and Methods

### 4.1. Chemicals

Acros Organics (USA) provided Heparin sodium, 3-amino-1-propanol (Ami), 1,4-diaminobutane (DAB), silver nitrate (AgNO_3_), 1-ethyl-3-3-dimethylaminopropyl carbodiimide (EDC), p-nitrophenyl chloroformate (NPC), N-hydroxysuccinimide (NHS). Sigma-Aldrich provided cis-Dichlorodiamineplatinum (II) (Cis), 99.99%, Curcuminoid (Cur), and poloxamer P403. Dialysis membranes (3.5 kDa and 12–14 kDa MW cut-off) were obtained from Repligen Corporation (Rancho Dominguez, Los Angeles, CA, USA).

### 4.2. Synthesis of Poloxamer P403-Conjugated Heparin Copolymers (HP403)

HP403 graft copolymer was prepared through the conjugation between a NPC-activated P403 (NPC-P403-OH) and aminated heparin (H-DAB), as shown in Figure 12.

#### 4.2.1. Synthesis of NPC-P403-OH

P403 (2.5 mmol) was activated with NPC (7.5 mmol) at 70 °C under vacuuming and stirring. After 8 h, we reduced the temperature back to R.T. and redissolved the sample in 20 mL THF before precipitating in a diethyl ether:hexane mixture (1:1 *v*/*v*), repeated three times. NPC-P403-NPC precipitant was obtained after rotary evaporation.

NPC-P403-NPC (1.5 mmol) was dissolved in THF (25 mL). Then, Ami (0.75 mmol in 25 mL THF) was drop-wised into the NPC-P403-NPC solution under stirring at R.T. for 12 h. NPC-P403-OH precipitant was obtained in viscous form after precipitation in a 1:1 (*v*/*v*) mixture of diethyl ether:hexane and rotary evaporation.

The chemical structure of NPC-P403-OH was analyzed via ^1^H-NMR spectrometry (Bruker, Billerica, MA, USA).

#### 4.2.2. Synthesis of H-DAB

DAB-aminated heparin derivatives (H:DAB = 1:3 mmol/mmol) with EDC/NHS as the carbodiimide coupling reagent.

#### 4.2.3. Synthesis of HP403 Copolymer

NPC-P403-OH was dissolved in water at 4 °C and slowly added to the H-DAB solution. The reaction lasted for 24 h at 15 °C. The sample was dialyzed with distilled water through the cellulose membrane (MWCO 12–14 kDa) for five days before freeze-drying. Product characteristics, structure, and morphology were determined using ^1^H-NMR and TEM. The critical micelle concentration (CMC) was investigated by the iodine probe method [49,59] and calculated via the piecewise function. The data were recorded three times.

### 4.3. Synthesis of HP403 Nanogel Coloading Curcumin and Cisplatin Hydrate (CisOH)

CisOH preparation referred to our previous reports [34]. In short, the reaction was carried out in an N_2_ atmosphere at room temperature with degassed DI water. The molar ratio of AgNO_3_ (0.88 mmol) and Cis (0.44 mmol) was 2:1. At 48 h after reaction, CisOH was obtained after centrifugation to remove the AgCl precipitate.

Cur (5 mg) was dissolved in 5 mL ethanol:dichloromethane (DCM) solvent at a ratio of 7:3 (*v*/*v*). HP403 (100 mg in 10 mL solution at 20 °C) under ultrasonic condition was added dropwise into Cur solution (30 min, temperature below 20 °C). The system was evaporated to remove solvent and dissolved by 5 mL of water. Then, the sample was centrifuged (15 min, 5000 rpm) and freeze-dried to obtain the HP403@Cur product (Figure 13).

CisOH (40 mg) solution was added dropwise into HP403@Cur products under constant magnetic stirring in a 20 °C nitrogen atmosphere for 24 h. Then, we dialyzed the achieved solution three times with distilled water at room temperature using cellulose membranes (MWCO 3.5 kDa, 20 min/each). HP403@CisOH@Cur product was obtained after freeze-drying (Figure 13). Product characteristics, structure, and morphology were analyzed by the Bruker AM500 FT NMR spectrometer (Bruker, Billerica, MA, USA) and TEM-1400 instrument (JEOL Ltd., Tokyo, Japan). The percentage of CisOH and Cur successfully co-loaded into HP403 nanogel was determined by inductively coupled plasma mass spestrometer (ICP-MS, NexION 2000, Perkin Elmer, MA, USA), and UV-Vis (Agilent, Santa Clara, CA, USA). Drug loading (DL%) and entrapment efficiency (EE%) of micelle polymer or nanogel were referred to published formulas [31,33,50]
(1)$$\mathrm{EE}\;\left(\%\right)=\frac{\mathrm{Weight}\;\mathrm{of}\;\mathrm{the}\;\mathrm{drug}\;\mathrm{in}\;\mathrm{micelles}}{\mathrm{Weight}\;\mathrm{of}\;\mathrm{the}\;\mathrm{feeding}\;\mathrm{drugs}}\times 100\%$$
(2)$$\mathrm{DL}\;\left(\%\right)=\frac{\mathrm{Weight}\;\mathrm{of}\;\mathrm{the}\;\mathrm{drug}\;\mathrm{in}\;\mathrm{micelles}}{\mathrm{Weight}\;\mathrm{of}\;\mathrm{copolymers}\;\mathrm{and}\;\mathrm{drugs}}\times 100\%$$

### 4.4. Release Profile and Kinetics of Drugs

#### 4.4.1. The Release Profile of Curcumin

HP403@CisOH@Cur in 2 mL DI water. The control sample was Cur reconstituted in 2 mL ethanol. Each solution was placed in a dialysis bag (MW cut off 3.5 kDa) and immersed in 20 mL PBS buffer at pH 5.5 and pH 7.4, respectively. Tween 80 was added into the dialysis environment to increase Cur dispersion and prevent its precipitation. The system was magnetically stirred at 100 rpm, 37 °C ± 1 °C. Cur content measurement was taken with 1 mL dialysis solution at given time points via an Agilent 8453 UV spectrophotometer (Agilent, Santa Clara, CA, USA). Simultaneously, 1 mL of PBS buffer containing Tween 80 was added to restore the initial volume. Cur was calculated as follows:(3)$$\mathrm{CR}\;\left(\%\right)=\sum_{t=0}^{t=\infty }\frac{M_{t}}{M_{0}}\times 100$$

Of which: Mt is the amount of Cur in the release environment at t (hours); M_0_ is the amount of Cur in HP403@CisOH@Cur. To investigate the Cur-released profile of HP403@CisOH@Cur, we analyzed obtained data with Korsmeyer–Peppas models, first-order, and zero-order. Regression results were indicated by the regression coefficient (R^2^) [60,61].

#### 4.4.2. The Release Profile of CisOH

The methods to study the CisOH release profile were similar to Cur. However, we prepared CisOH control samples with 2 mL DI water. CisOH contents were determined by platinum measurements via ICP-MS (NexION 2000, Perkin Elmer, MA, USA) (inductively coupled plasma mass spestrometer) and AOAC (association of analytical communities). The CisOH release experiments were also repeated 3 times.

#### 4.4.3. Stability Test of HP403@CisOH@Cur Nanogel

HP403@CisOH@Cur was dispersed in DI water after 96 h storage at 37 °C. All tests were performed at 37 °C by DLS HORIBA SZ-100 nanoparticle size analyzer (Horiba Ltd., Kyoto, Japan) to investigate HP403@CisOH@Cur’s stability at 0, 12, 24, 48, 72 and 96 h at 37 °C. Experiments were measured in triplicate for each sample, and the results are expressed as mean ± SD.

### 4.5. In Vitro Cytotoxic Assay on MCF-7 Cell Line

The cytotoxicity assay of HP403, HP403@CisOH and HP403@CisOH@Cur was performed on the breast cancer cell line MCF-7 using SRB assay. [62]. The experiments were conducted at the Molecular Biology Laboratory of the University of Science, Vietnam National University, Ho Chi Minh City, Vietnam. Briefly, 5 × 10^3^ MCF-7 cells were incubated each well in 96-well plates. The IC_50_ value, inhibitory activity of free Cis, HP403@CisOH, and HP403@CisOH@Cur against MCF-7 proliferation were calculated and compared among samples.

### 4.6. In Vivo Anti-Tumor Activity

Male white mice (*Mus musculus* var. Albino), at 6–7 weeks old and weighing 25–30 g, were provided by the Stem Cell Institute, University of Sciences, Vietnam National University, HCMC. Mice were raised in laboratory conditions of 20–30 °C, humidity of 30–50% under a 12 h–light–dark photoperiod.

Methods of creating an immunodeficiency mouse model and mouse model with heterogeneous tumors followed the Department of Animal Physiology and Biotechnology’s procedure, University of Sciences, Vietnam National University, HCMC. Animal experiments described in this study were performed in compliance with institutional guidelines and according to protocol approved by the Animal Care and Use Committee of the University of Science under Ethics approval number 500B/KHTN-ACUCUS (9 June 2021).

The total of 25 mice that successfully carried tumors was randomly divided into 5 treatments (5 mice each treatment): physiological saline NaCl 0.9% (100 µL/dose), HP403 (12 mg per kg body weight), Cis (3 mg per kg body weight), HP403@CisOH (3 mg Cis per kg body weight), HP403@CisOH@Cur (3 mg Cis per kg body weight, and Cur: HP403 = 5:100 wt/wt).

The drug was injected through tail veins every 3 days for 13 days with each dose of 3 mg Cis per kg body weight. Body weight and tumor size were measured daily. The largest and smallest diameter of the tumor was recorded to calculate the tumor volume. At the end of 13 days of the drug trial, tumors were collected and preserved in formalin 10% solution, then underwent hematoxylin and eosin staining to observe density of tumor cells after various treatments.

### 4.7. Data Analysis

ORIGIN 8.5.1 (OriginLab Inc., Northampton, MA, USA) was used to perform statistical analysis. All experiments were carried out for at least three independent replications and data were expressed as the mean ± SD. One-way ANOVA or two-way ANOVA tests and Tukey multiple comparison tests were used to determine statistical differences. A * *p* < 0.05, ** *p* < 0.01, *** *p* < 0.001 indicated statistical significance, while ns *p* ≥ 0.05 meant that the difference was not statistically significant.

The Materials and Methods should be described with sufficient details to allow others to replicate and build on the published results. Please note that the publication of your manuscript implicates that you must make all materials, data, computer code, and protocols associated with the publication available to readers. Please disclose at the submission stage any restrictions on the availability of materials or information. New methods and protocols should be described in detail while well-established methods can be briefly described and appropriately cited.

Research manuscripts reporting large datasets that are deposited in a publicly available database should specify where the data have been deposited and provide the relevant accession numbers. If the accession numbers have not yet been obtained at the time of submission, please state that they will be provided during review. They must be provided prior to publication.

Interventionary studies involving animals or humans, and other studies that require ethical approval, must list the authority that provided approval and the corresponding ethical approval code.

## Figures and Tables

**Figure 1 gels-08-00059-f001:**
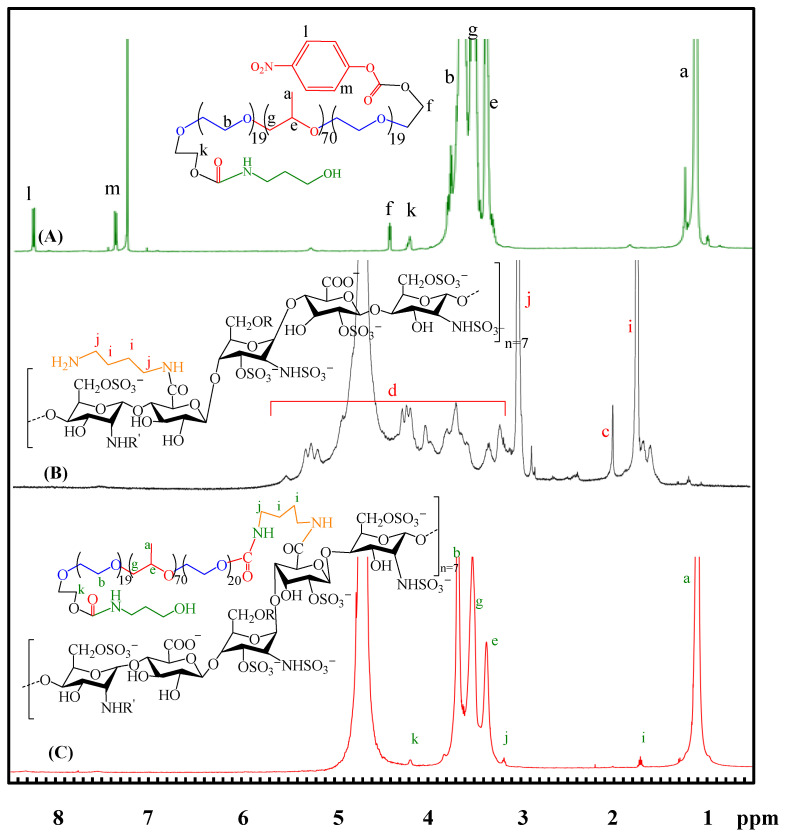
^1^H NMR spectra of NPC-P403-OH (**A**); H-DAB (**B**); HP403 (**C**).

**Figure 2 gels-08-00059-f002:**
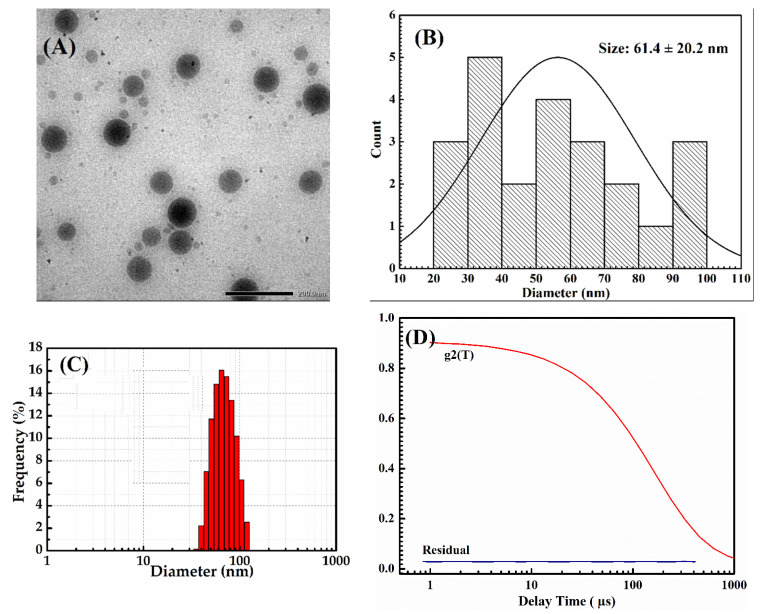
TEM image (**A**), diameter histogram (**B**), DLS particle size distribution (**C**), and stability (**D**) of the HP403 nanogel.

**Figure 3 gels-08-00059-f003:**
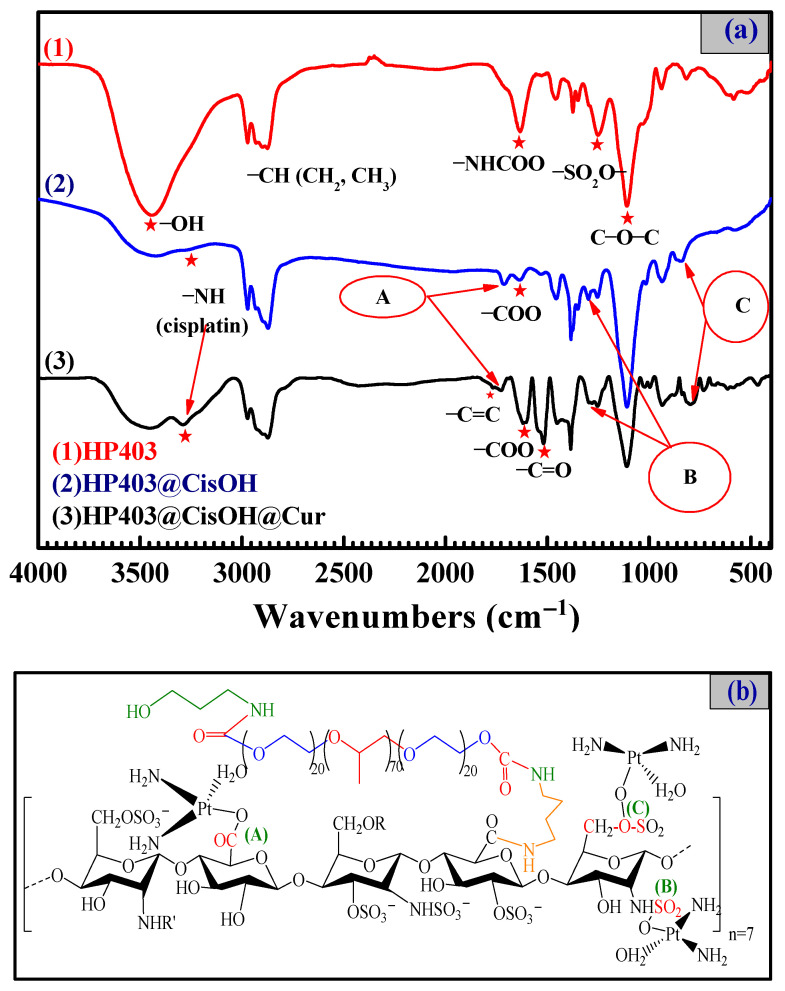
FT-IR spectra of heparin, HP403, HP403@CisOH and HP403@CisOH @Cur (**a**); Simple illustration of HP403@CisOH complex (**b**).

**Figure 4 gels-08-00059-f004:**
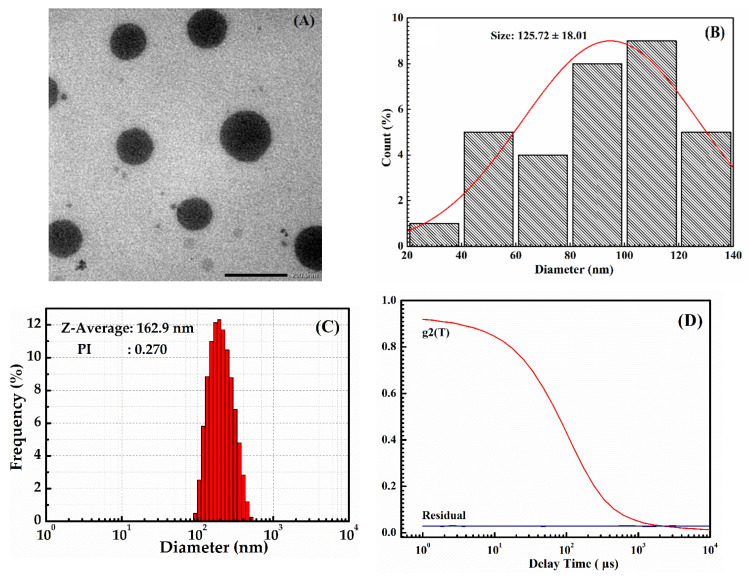
TEM image (**A**), diameter histogram (**B**), DLS particle size distribution (**C**), and stability (**D**) of the HP403@CisOH@Cur nanogel.

**Figure 5 gels-08-00059-f005:**
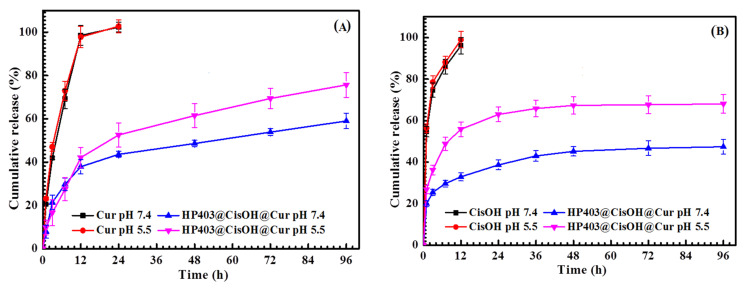
Released profiles of Cur (**A**) and CisOH (**B**) from the HP403@CisOH@Cur drug-loading system and control samples at physiological conditions, pH 5.5 and pH 7.4 (37 ± 1 °C; n = 3).

**Figure 6 gels-08-00059-f006:**
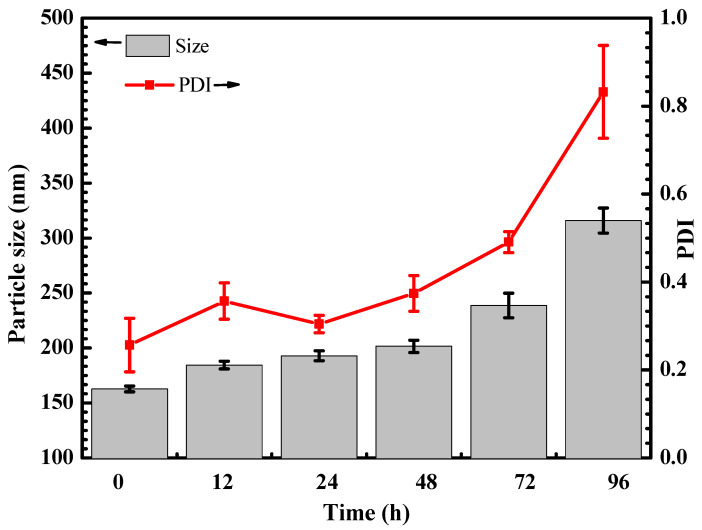
Particle size and PDI of HP403@CisOH@Cur over time after redispersion in an aqueous medium at 37 °C.

**Figure 7 gels-08-00059-f007:**
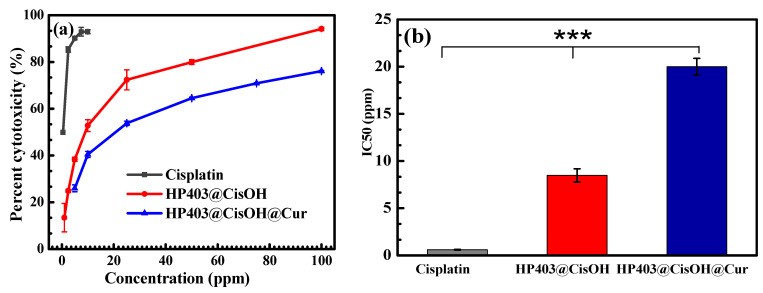
Cytotoxicity of Cis, HP403@CisOH and HP403@CisOH@Cur on MCF-7 cells, in terms of (**a**) Cell death percentage and (**b**) IC50 value. (Statistical significance of *p* < 0.001 are indicated by ***).

**Figure 8 gels-08-00059-f008:**
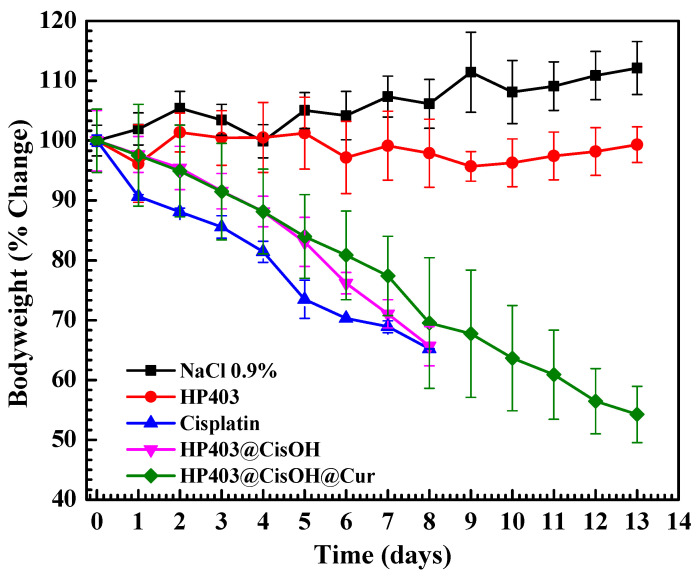
Change in body weight of mice after various formular injections.

**Figure 9 gels-08-00059-f009:**
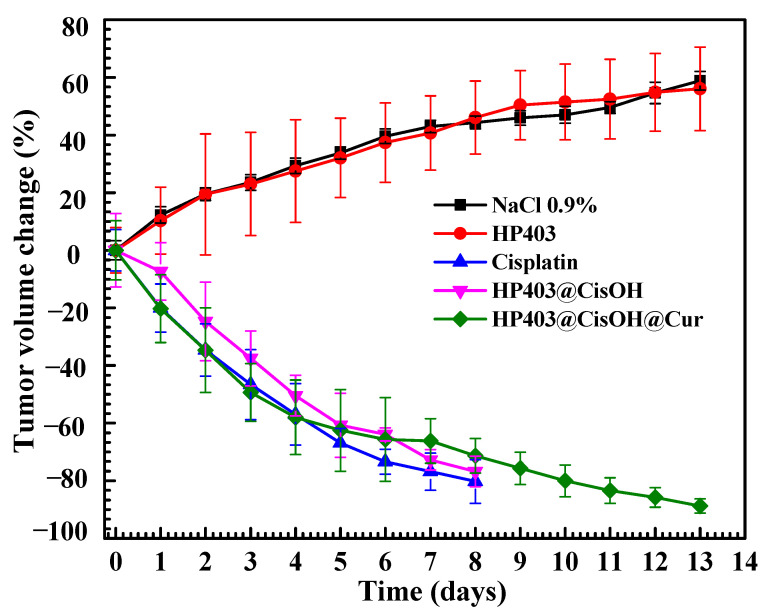
Change in tumor volume of mice after various formular injections.

**Figure 10 gels-08-00059-f010:**
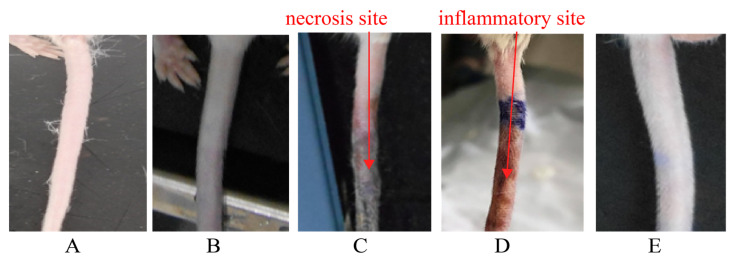
Tail at injection site of treatments (**A**) NaCl (**B**) HP403 (**C**) Cis (**D**) HP403@CisOH (**E**) HP403@CisOH@Cur.

**Figure 11 gels-08-00059-f011:**
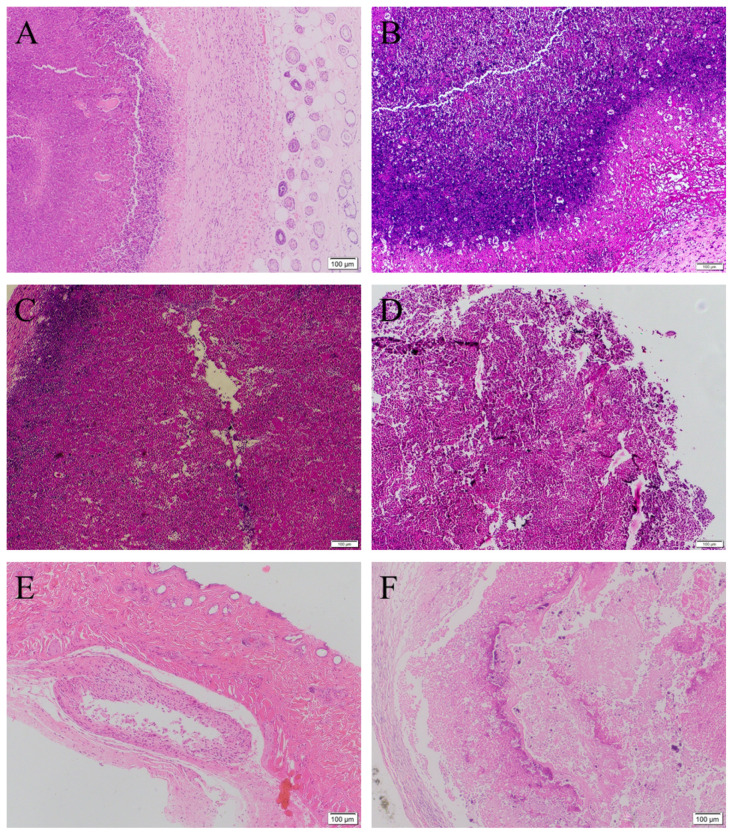
Histological H&E images of tumor tissues before (**A**) and after treatment with NaCl (**B**), HP403 (**C**), Cis (**D**), HP403@CisOH (**E**), and HP403@CisOH@Cur (**F**).

**Figure 12 gels-08-00059-f012:**
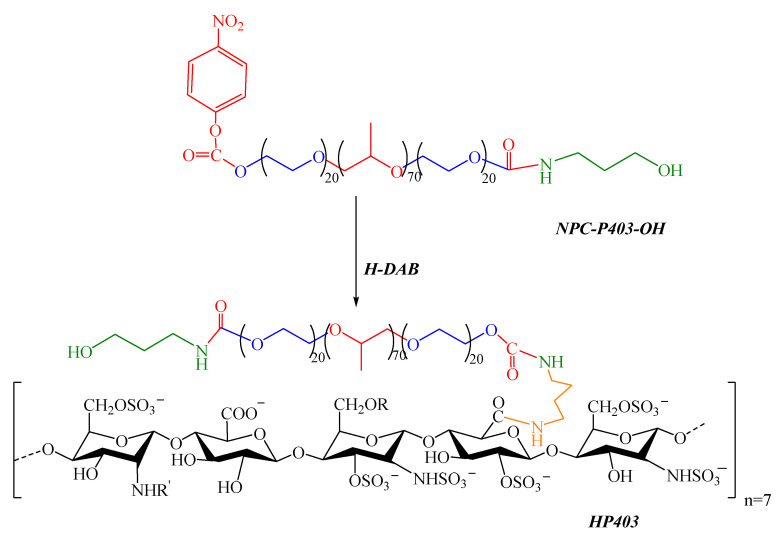
Synthetic scheme of the HP403 graft copolymer (R, R=H, SO_3_^−^ or COCH_3_).

**Figure 13 gels-08-00059-f013:**
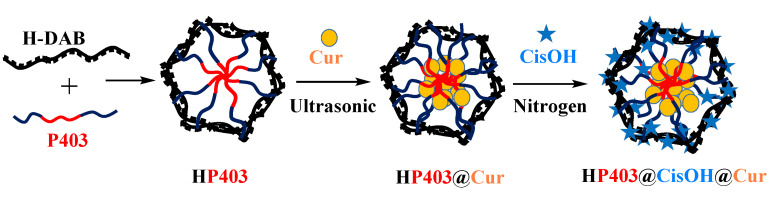
Schematic diagram of HP403 nanogel; HP403@Cur and HP403@CisOH@Cur.

**Table 1 gels-08-00059-t001:** Release parameters for Cur and CisOH of HP403@CisOH@Cur were obtained after fitting the drug-release data to four different mathematical models of drug release kinetics.

Cur Formulation	pH	Mathematical Models for Drug-Release Kinetics								
		Zero Order		First Order		Higuchi		Power Law		
		*k* _0_	*R* ^2^	*k* _1_	*R* ^2^	*k_H_*	*R* ^2^	*K*	*n*	*R* ^2^
Free Cur	pH 5.5	0.5537	0.8730	0.1318	0.8011	1.1075	0.8730	3.1978	0.3933	0.9629
HP403@CisOH@Cur	pH 5.5	0.0986	0.8957	0.0278	0.7948	0.1972	0.9097	1.2726	0.4225	0.9801
Free Cur	pH 7.4	1.2622	0.8779	0.1307	0.8104	1.0962	0.8779	2.8779	0.4271	0.9590
HP403@CisOH@Cur	pH 7.4	0.0782	0.8430	0.0245	0.8104	0.1563	0.8430	1.5877	0.2929	0.9748
**Cis Formulation**	**pH**	**Mathematical Models for Drug-Release Kinetics**								
		**Zero Order**		**First Order**		**Higuchi**		**Power Law**		
		** *k* _0_ **	** *R* ^2^ **	** *k* _1_ **	** *R* ^2^ **	** *k_H_* **	** *R* ^2^ **	** *k* **	** *n* **	** *R* ^2^ **
Free CisOH	pH 5.5	1.0435	0.9746	0.2614	0.9569	2.0870	0.9746	5.2155	0.2433	0.9880
HP403@CisOH@Cur	pH 5.5	0.1047	0.7598	0.0395	0.7519	1.0624	0.9141	2.7966	0.2646	0.9579
Free CisOH	pH 7.4	1.0797	0.9719	0.2657	0.9498	2.1594	0.9719	5.4818	0.2359	0.9992
HP403@CisOH@Cur	pH 7.4	0.0706	0.8752	0.0307	0.8641	0.6281	0.9615	1.9985	0.2056	0.9965

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
