# Peer review of "Curcuminoid Co-Loading Platinum Heparin-Poloxamer P403 Nanogel Increasing Effectiveness in Antitumor Activity"

_gels, 2022, doi:10.3390/gels8010059_

Round 1

Reviewer 1 Report

The manuscript by Ngoc The Nguyen et al. introduces a new type of drug carrier that reduced the side effects of platinum drugs, and enhanced the anti-tumor effect through the synergistic effect of platinum compounds and curcumin, providing new ideas for the optimization of chemotherapy. But the experimental results are not ideal, some changes are needed. Below my suggestions to improve the manuscript:

- The experimental results showed that the weight of mice in the HP403@CisOH@Cur group dropped to about 50% at 13 days, indicating that the drug is highly toxic and has little clinical application value.

- This paper is not concise enough and lacks highlights. It is recommended to adjust the introduction part, first introduce the two drugs and then introduce the carrier. In addition, the results are not fully discussed, and it is recommended to extend the comparison with other similar published work.

- There are some problems in the article’s figures. For example, How figure 3A quantified the DL and EE results of HP403@CisOH@Curit;  Figure 5A does not reflect that Cur released more efficiently at pH 5.5; Why do all three pillars have asterisks in figure 6b.

- Some abbreviations were not explained when they first appeared. For example, HP403@CisOH@Cur in line 27 on page 1, NPC in line 92 and DAB in line 97 on page 3.

- The reference format is a bit messy, please consult the journal’s reference style for the exact appearance of these elements.

- Please add a schematic diagram of the nanocarrier structure and a schematic diagram of the drug loading process, so that readers can quickly understand the content of the paper.

Author Response

Dear - Editor-in-Chief of the Gels

         - Response to Reviewer 1

Thank you very much for processing our manuscript.

We have revised the manuscript according to comments from reviewer, answers reviewer’s opinions and queries are presented in attached file.

Reviewer 2 Report

  1. Figure 10 A, B: Two photos of the same images.
  2. Stability issues of the formulation are required to be addressed.
  3. Table 1 haas not been discussed adequately. What is the possible mechanism of drug release.
  4. Approved animal study protocol is required.
  5. Discuss is required to be more conclusive.
  6. Check overall english of the manuscript.

Author Response

Dear - Editor-in-Chief of the Gels

         - Response to Reviewer 2

Thank you very much for processing our manuscript.

We have revised the manuscript according to comments from reviewer, answers reviewer’s opinions and queries are presented in attached file.

Round 2

Reviewer 1 Report

Compared with the previous edition, this paper has been improved obviously. 

I have another small question about Figure 13, why is CisOH attached to the surface of the nanocarrier, instead of being contained in it  like Cur?

Author Response

Dear Reviewer,

Thank you very much for processing our manuscript.

We have responded to Reviewer comment, please see the attached file as below.

Kind regards,

Dr. The Nguyen

Reviewer 2 Report

Animal study protocol with approval number is required. Kindly provide all the documents pertaining to the animal study protocol (minutes of the meeting, protocol, protocol presentation etc)

Author Response

Dear Reviewer,

Thank you very much for processing our manuscript.

We have revised the manuscript according to comments from reviewer, please see the attached file as below.

Kind regards,

Dr. The Nguyen

Round 3

Reviewer 2 Report

The manuscript has been significantly revised.

All the concerns have been adequately addressed.